# Potential impact of outpatient stewardship interventions on antibiotic exposures of common bacterial pathogens

Christine Tedijanto[1]*, Yonatan H Grad[2,3], Marc Lipsitch[1,2]

[1]Center for Communicable Disease Dynamics, Department of Epidemiology, Harvard T.H. Chan School of Public Health, Boston, United States; [2]Department of Immunology and Infectious Diseases, Harvard T.H. Chan School of Public Health, Boston, United States; [3]Division of Infectious Diseases, Brigham and Women's Hospital, Harvard Medical School, Boston, United States

**Abstract** The relationship between antibiotic stewardship and population levels of antibiotic resistance remains unclear. In order to better understand shifts in selective pressure due to stewardship, we use publicly available data to estimate the effect of changes in prescribing on exposures to frequently used antibiotics experienced by potentially pathogenic bacteria that are asymptomatically colonizing the microbiome. We quantify this impact under four hypothetical stewardship strategies. In one scenario, we estimate that elimination of all unnecessary outpatient antibiotic use could avert 6% to 48% (IQR: 17% to 31%) of exposures across pairwise combinations of sixteen common antibiotics and nine bacterial pathogens. All scenarios demonstrate that stewardship interventions, facilitated by changes in clinician behavior and improved diagnostics, have the opportunity to broadly reduce antibiotic exposures across a range of potential pathogens. Concurrent approaches, such as vaccines aiming to reduce infection incidence, are needed to further decrease exposures occurring in 'necessary' contexts.

*For correspondence:
ctedijanto@g.harvard.edu

## Introduction

Antibiotic consumption is a known driver of antibiotic resistance. In developed nations, over 80% of antibiotic consumption for human health occurs in the outpatient setting (*European Centre for Disease Prevention and Control, 2018*; *Public Health Agency of Canada, 2018*; *Public Health England, 2017*; *Swedres-Svarm, 2017*), and US-based studies conducted across different subpopulations have estimated that 23–40% of outpatient prescriptions may be inappropriate (*Chua et al., 2019*; *Fleming-Dutra et al., 2016*; *Olesen et al., 2018b*). Inappropriate antibiotic use leads to increased risk of adverse events (*Linder, 2008*; *Shehab et al., 2008*), disruption of colonization resistance and other benefits of the microbial flora (*Buffie and Pamer, 2013*; *Khosravi and Mazmanian, 2013*), and bystander selection for antibiotic resistance, with little to no health gains for the patient. Recent work on bystander selection estimates that, for 8 of 9 potential pathogens of interest, over 80% of their exposures to commonly used antibiotic classes in the outpatient setting occur when the organisms are asymptomatically colonizing the microbiome, not causing disease (*Tedijanto et al., 2018*). A corollary of extensive bystander selection is that reductions in use will prevent antibiotic exposures for species throughout the microbiome.

We use publicly available data to quantify potentially avertable exposures of bacterial species to commonly used antibiotics under hypothetical changes in prescribing practice. The set of scenarios included here are intended as thought experiments to explore the upper bounds of avertable

antibiotic exposures. Reductions in antibiotic consumption have historically had variable impacts on resistance levels, likely dependent on setting, baseline consumption and resistance patterns, and fitness costs (*Andersson and Hughes, 2010*; *Lipsitch, 2001*). In addition, treatment strategies are primarily guided by randomized controlled trials assessing immediate clinical outcomes, with less careful consideration given to the microbiome-wide effect of such decisions. We apply analytical methods to parse out species-level effects of changes in prescribing practice in order to better understand the potential impact of antibiotic stewardship.

## Materials and methods

### Scenarios of interest

This analysis includes sixteen antibiotics that are frequently prescribed in the outpatient setting and nine potentially pathogenic bacterial species that are commonly carried in the normal human microbiome. For each antibiotic-species pair, we estimate the proportion of antibiotic exposures experienced by that species that could be averted under four hypothetical scenarios. The scenarios range from broad elimination of unnecessary prescribing to focused modifications of antibiotic use for specific indications as follows:

1. Eliminate unnecessary antibiotic use across all outpatient conditions.
2. Eliminate all antibiotic use for outpatient respiratory conditions for which antibiotics are not indicated.
3. Eliminate all antibiotic use for acute sinusitis.
4. Prescribe nitrofurantoin for all cases of cystitis in women.

We use the results reported by Fleming-Dutra and colleagues as estimates of unnecessary antibiotic use in ambulatory care settings (*Fleming-Dutra et al., 2016*). In this study, the authors convened a group of experts to estimate the proportion of appropriate antibiotic use by condition and age group based on clinical guidelines. When guidelines could not be used for this task, the rate of appropriate antibiotic prescribing was estimated by benchmarking against the lowest-prescribing US region. Based on the same paper, we consider viral upper respiratory tract infection, influenza, non-suppurative otitis media, viral pneumonia, bronchitis, and allergy and asthma to be outpatient respiratory conditions for which antibiotics are not indicated.

Antibiotic treatment of acute bacterial sinusitis is currently guideline-recommended (*Rosenfeld et al., 2015*). However, bacteria are an infrequent cause of acute sinusitis, and due to the self-limiting nature of the syndrome, evidence to support antibiotic treatment is weak (*Burgstaller et al., 2016*; *Fokkens et al., 2007*). The cause of sinusitis (whether bacterial, viral, or noninfectious) can be very difficult to distinguish in practice, and as a result of this challenge and others, antibiotics continue to be prescribed at over 80% of US outpatient visits with a primary diagnosis of acute sinusitis (*Smith et al., 2013*). In contrast, antibiotics are always recommended for urinary tract infections (UTIs). Despite being recommended as a second-line therapy for cystitis due to concerns about resistance, fluoroquinolones are the most common treatment, prescribed at 40% of outpatient visits for uncomplicated UTI (*Kabbani et al., 2018*). We explore the hypothetical scenario of treating all cases of cystitis in women with nitrofurantoin, a recommended first-line therapy with good potency against common uropathogens, low levels of resistance, and decreased risk of collateral damage to the intestinal microbiome due to its propensity to concentrate in the bladder (*Gupta et al., 2011*; *Gupta et al., 1999*; *Stewardson et al., 2015*). Nitrofurantoin is considered clinically appropriate unless the patient has chronic kidney disease, is showing signs of early pyelonephritis, or has had a prior urinary isolate resistant to nitrofurantoin (*Hooton and Gupta, 2019*). In our analysis, we exclude patients with a simultaneous diagnosis of pyelonephritis. We assume that other contraindications are fairly rare in the general population.

### Data sources and methodology

Extending our recent work on bystander selection (*Tedijanto et al., 2018*), we used the 2015 National Ambulatory Medical Care Survey and National Hospital Ambulatory Medical Care Survey (NAMCS/NHAMCS) (*National Center for Health Statistics, 2015*), the Human Microbiome Project (HMP) (*Huttenhower et al., 2012*; *Human Microbiome Project Consortium, 2012*), and assorted

carriage and etiology studies (*Bäckhed et al., 2015*; *Bluestone et al., 1992*; *Bogaert et al., 2011*; *Brook et al., 1994*; *Celin et al., 1991*; *Chira and Miller, 2010*; *Edlin et al., 2013*; *Ginsburg et al., 1985*; *Gunnarsson et al., 1997*; *Gupta et al., 1999*; *Gwaltney et al., 1992*; *Hammitt et al., 2006*; *Lif Holgerson et al., 2015*; *Huang et al., 2009*; *Jain et al., 2015*; *Mainous et al., 2006*; *Pettigrew et al., 2012*; *Regev-Yochay et al., 2004*; *Verhaegh et al., 2010*; *Wubbel et al., 1999*; *Yassour et al., 2016*) (details in *Figure 1—source data 1*) to estimate national outpatient antibiotic exposures by drug, species, and condition. NAMCS/NHAMCS are annual cross-sectional surveys designed to sample outpatient visits in the United States, and up to five diagnoses and thirty medications may be associated with each visit. We used methodology developed by Fleming-Dutra and colleagues (*Fleming-Dutra et al., 2016*), and applied in other studies (*Olesen et al., 2018a*), to group diagnosis codes into conditions and link antibiotic prescriptions with the most likely indication. For this analysis, visits with a diagnosis of acute cystitis (ICD-9-CM: 595.0), unspecified cystitis (595.9) or unspecified UTI (599.0), without a concurrent diagnosis of pyelonephritis (590.1, 590.8), renal abscess (590.2), or kidney infection (590.9), were considered to be associated with cystitis. In this analysis, we maintain the assumption that one antibiotic prescription is equivalent to one exposure; antibiotic exposures experienced by a given species and associated with a given condition are roughly estimated as the product of antibiotic prescriptions for that condition and species carriage prevalence, which is dependent on disease etiology (target exposures) and asymptomatic carriage prevalence (bystander exposures). For diagnoses where etiology was not readily available, we assumed that none of our species of interest were causative agents. These assumptions have been enumerated in detail in prior work (*Tedijanto et al., 2018*). We applied proportions of unnecessary antibiotic prescribing by condition and age group estimated by Fleming-Dutra and colleagues based on expert opinion, clinical guidelines, and regional variability in use (*Fleming-Dutra et al., 2016*). For relevant scenarios (1 and 2), we applied the proportions of unnecessary use evenly across all antibiotic prescriptions. Antibiotics and antibiotic classes are identified by the Lexicon Plus, 2008 classification scheme (https://www.cerner.com/solutions/drug-database).

The proportion of avertable antibiotic exposures for each species is defined by *Equation 1*. The equation adopts previously described notation (*Tedijanto et al., 2018*) with modifications. A listing of all variables and descriptions can be found in *Table 1*. Let $a$ represent antibiotic, $s$ represent species, $i$ represent ICD-9-CM diagnosis code, and $g$ represent age group. Throughout the analysis we have weighted outpatient visits to be nationally representative using the sampling and nonresponse weights provided in NAMCS/NHAMCS. Let $X_{as}$ be the number of avertable exposures by antibiotic and species, $T_{as}$ be the total number of exposures by antibiotic and species, $d_{aig}$ be the number of prescriptions of antibiotic $a$ associated with diagnosis code $i$ in age group $g$, $p_{sig}$ be the carriage prevalence of species $s$ among those with diagnosis code $i$ in age group $g$, and $q_{aig}$ be the proportion of avertable exposures by diagnosis code and age group in the given scenario. For example, in the scenario assessing elimination of all unnecessary antibiotic use, $q_{aig}$ is the proportion of avertable

**Table 1.** Notation, descriptions, and sources for variables in *Equation 1*.

| Variable | Definition | Source |
|---|---|---|
| $d_{aig}$ | Number of prescriptions (using nationally representative weights) of antibiotic $a$ associated with ICD9-CM diagnosis code $i$ in age group $g$ | NAMCS/NHAMCS 2015 |
| $e_{sig}$ | Proportion of cases of condition defined by ICD9-CM diagnosis code $i$ in age group $g$ caused by species $s$ | Published etiology studies (see *Figure 1—source data 1*) |
| $p_{s0g}$ | Asymptomatic carriage prevalence of species $s$ in age group $g$ | Human Microbiome Project (HMP) and published carriage studies |
| $p_{sig}$ | Carriage prevalence of species $s$ among individuals diagnosed with ICD9-CM code $i$ in age group $g$ | $e_{sig} + (1 - e_{sig})p_{s0g}$ |
| $q_{aig}$ | Proportion of prescriptions of antibiotic $a$ associated with ICD9-CM diagnosis code $i$ in age group $g$ that are avertable under the given scenario | Based on article by Fleming-Dutra et al. (*Fleming-Dutra et al., 2016*) with adjustments as described in *Figure 1—source data 2* |
| $X_{as}$ | Number of exposures of antibiotic $a$ experienced by species $s$ that are avertable under the given scenario | $\sum_{g=1}^{G} \sum_{i=0}^{I} d_{aig} \times p_{sig} \times q_{aig}$ |
| $T_{as}$ | Total number of exposures of antibiotic $a$ experienced by species $s$ | $\sum_{g=1}^{G} \sum_{i=0}^{I} d_{aig} \times p_{sig}$ |

antibiotic use by diagnosis and age group (*Fleming-Dutra et al., 2016*). Alternatively, in the scenario assessing elimination of non-nitrofurantoin treatment for cystitis, $q_{aig}$ is 1 when $a$ is not nitrofurantoin, $i$ is a diagnosis code associated with cystitis, and the patient is female, and 0 elsewhere. Carriage prevalences ($p_{sig}$) are assumed to be constant within three age groups (under 1 year, 1–5 years, over 5 years old) (*Tedijanto et al., 2018*), while proportions of avertable antibiotic use ($q_{aig}$) were reported for three age groups (0–19 years, 20–64 years, 65 years old and over) (*Fleming-Dutra et al., 2016*). $G$ is the smallest set of age groups that accounts for this granularity (under 1 year, 1–5 years, 6–19 years, 20–64 years, 65 years old and over). For antibiotic prescriptions that occurred at visits without any ICD-9-CM diagnosis codes ($i = 0$), we applied the carriage prevalence among healthy individuals.

*Equation 1*. Proportion of avertable exposures by species and antibiotic.

$$\frac{X_{as}}{T_{as}} = \frac{\sum\limits_{g=1}^{G}\sum\limits_{i=0}^{I} d_{aig} \times p_{sig} \times q_{aig}}{\sum\limits_{g=1}^{G}\sum\limits_{i=0}^{I} d_{aig} \times p_{sig}}$$

In scenarios where unnecessary use is eliminated (1 and 2), we assume that only bystander exposures are affected. This presumes that perfect discrimination between bacterial and non-bacterial etiologies is possible. This results in a slight modification to the numerator of *Equation 1* – $p_{sig}$ is changed to $p_{s0g}$ as all eliminated exposures would have occurred during asymptomatic carriage. For cases where $e_{sig}$ is 1, we assume zero avertable exposures. In addition, we make slight modifications to $q_{aig}$ when we estimate the proportion of bacterial cases to be larger than the proportion of necessary prescriptions (*Figure 1—source data 2*; *Bluestone et al., 1992*; *Brook et al., 1994*; *Celin et al., 1991*; *Brook, 2016*). As a sensitivity analysis, we include the proportion of avertable exposures for each antibiotic-species pair under Scenario 1 if antibiotic use was eliminated equally across both target and bystander exposures (*Figure 1—figure supplement 1*). All analysis was conducted in R version 3.6.1.

## Results

### Results for all four scenarios are depicted in *Figure 1*

#### Scenario 1
We estimate that elimination of unnecessary antibiotic prescriptions across all outpatient conditions would prevent 6% to 48% (IQR: 17% to 31%) of antibiotic-species exposures (*Figure 1A*). The smallest reduction is associated with *S. pyogenes* exposures to cefdinir and the largest with *H. influenzae*, *E. coli,* and *P. aeruginosa* exposures to azithromycin. If all unnecessary antibiotic use could be prevented, over 30% of exposures to amoxicillin-clavulanate, penicillin, azithromycin, clarithromycin, levofloxacin, and doxycycline across most potential pathogens of interest could be averted. Of particular interest, elimination of all unnecessary prescribing in the outpatient setting could reduce exposures of *S. pneumoniae* to penicillins and macrolides by 27% and 37%, respectively, and of *S. aureus* to penicillins and quinolones by 27% and 21%. For *E. coli* and *K. pneumoniae*, approximately one-quarter of exposures to cephalosporins and one-fifth of exposures to quinolones could be averted.

#### Scenario 2
Scenario 2, elimination of all antibiotic use for outpatient respiratory conditions where antibiotics are not indicated, is a subset of Scenario 1 and accounts for a substantial portion of avertable exposures included in the first scenario. As in Scenario 1, the results are primarily driven by drug and depend on the amount of use of that drug for the affected conditions (in this case, conditions for which antibiotics are not indicated). Pathogen characteristics affecting the proportion of avertable exposures can be better understood by looking across all species for a single drug. For example, we focus on the row for azithromycin. Based on NAMCS/NHAMCS 2015, 32% of all azithromycin use is associated with the affected conditions. Since these conditions are never caused by our organisms of interest, the maximum proportion of avertable exposures for any species is 32%. However, the

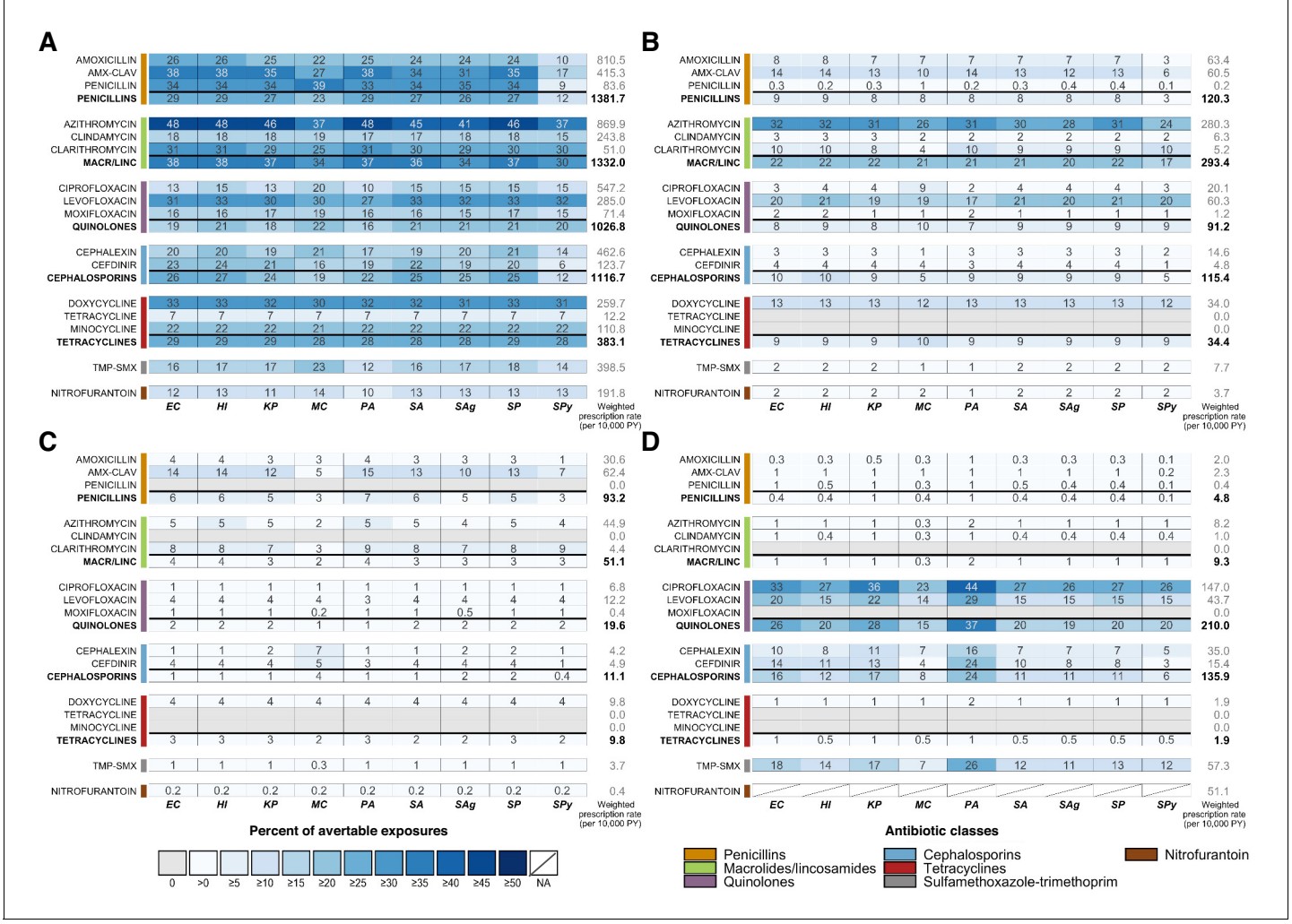

**Figure 1.** Heatmaps showing the estimated percentage of species exposures to each antibiotic or antibiotic class that could be averted by scenario. Scenarios are defined as elimination of (**A**) unnecessary antibiotic prescriptions across all outpatient conditions, (**B**) all antibiotic use for outpatient respiratory conditions for which antibiotics are not indicated, (**C**) all antibiotic use for acute sinusitis, and (**D**) non-nitrofurantoin treatment of cystitis in women. Drug class results include prescriptions of all antibiotics in that class, as identified by the Lexicon Plus classification system. Sensitivity and other additional analyses are shown in *Figure 1—figure supplements 1–3*. Abbreviations: *Antibiotics (y-axis):* AMX-CLAV = amoxicillin-clavulanate, MACR/LINC = macrolides/ lincosamides, TMP-SMX = sulfamethoxazole-trimethoprim; *Organisms (x-axis):* EC = *E. coli*, HI = *H. influenzae*, KP = *K. pneumoniae*, MC = *M. catarrhalis*, PA = *P. aeruginosa*, SA = *S. aureus*, SAg = *S. agalactiae*, SP = *S. pneumoniae*, SPy = *S. pyogenes*; PY = person years.

The online version of this article includes the following source data and figure supplement(s) for figure 1:

**Source data 1.** Summary of etiology studies.

**Source data 2.** Modifications to proportions of unnecessary antibiotic prescriptions based on published etiology studies and antibiotic use in NAMCS/NHAMCS.

**Source data 3.** Estimated antibiotic class exposures by exposed and causative species.

**Figure supplement 1.** Sensitivity analysis of proportions of avertable exposures across all outpatient conditions when proportion of unnecessary use is applied equally across target and bystander exposures (*Equation 1* without modification).

**Figure supplement 2.** Sensitivity analyses for Scenario 3 (elimination of all antibiotic use for acute sinusitis).

**Figure supplement 3.** Avertable exposures under Scenario 1 (elimination of all unnecessary antibiotic use across outpatient conditions) using 2010-2011 NAMCS/NHAMCS data with (left) and without (right) NHAMCS outpatient department data.

proportion of avertable exposures is modulated by the number of azithromycin exposures that occur for treatment of other conditions – for example, a lower proportion of avertable exposures for *S. pyogenes*, *M. catarrhalis*, and *S. agalactiae* indicates that they have relatively frequent exposure to

azithromycin during treatment of other conditions that are unaffected in this scenario, both as a bystander and causative pathogen.

## Scenario 3

In Scenario 3, we explore elimination of all antibiotic use for acute sinusitis. Overall, acute sinusitis accounts for just 3% of prescriptions of our included antibiotic classes, resulting in the low proportions of avertable exposures across most bacteria-antibiotic combinations. However, proportions of avertable exposures are relatively high for amoxicillin-clavulanate and clarithromycin, antibiotics for which a large proportion of their use is due to acute sinusitis (15% and 9%, respectively). In this scenario, we make the assumption that all use for acute sinusitis is unnecessary. In reality, a small number of cases are truly bacterial and antibiotics are indicated for patients with persistent, severe or worsening symptoms (*Rosenfeld et al., 2015*). We explore varying levels of unnecessary use based on etiological data in *Figure 1—figure supplement 2* (*Aitken and Taylor, 1998*; *Benninger et al., 2000*; *Fleming-Dutra et al., 2016*; *Sande and Gwaltney, 2004*; *Snow et al., 2001*; *Wald et al., 1991*; *Williams et al., 1992*).

## Scenario 4

In this scenario, we consider a program to prescribe only nitrofurantoin for all cases of acute cystitis in women, replacing all other antibiotics that are currently in use. As a result of this change, we estimate that exposures of *E. coli* and *K. pneumoniae* to cephalosporins could be reduced by 16% and 17%, respectively, and exposures to quinolones could decrease by 26% and 28%. Uniquely in this scenario among those we considered, several of our species of interest cause a large proportion of UTI cases (*E. coli*, *K. pneumoniae*, and *P. aeruginosa*). To explore the effects of being a causative pathogen on the proportion of avertable exposures, we focus on the results for ciprofloxacin. Overall, 27% of ciprofloxacin use is associated with acute cystitis in women. Organisms that are not causative pathogens of cystitis have proportions of avertable exposures close to or less than 27%, but a substantially higher proportion of exposures could be averted among causative pathogens - 33%, 36%, and 44% of ciprofloxacin exposures to *E. coli*, *K. pneumoniae*, and *P. aeruginosa*, respectively. Although *E. coli* is the most common cause of UTIs, its estimated proportion of avertable exposures in this scenario is lower than that of *P. aeruginosa*, likely due to the larger number of antibiotic exposures incurred by *E. coli* during asymptomatic carriage.

This scenario also allows us to observe the impact of differences in disease incidence, antibiotic use patterns, and carriage prevalence across age groups on the proportion of avertable exposures. For example, *M. catarrhalis* is asymptomatically carried in children much more frequently than in adults. Since cystitis occurs primarily in adults, the proportion of avertable exposures is often lower for *M. catarrhalis* compared to other species due to bystander exposures in children that are largely unaffected by modified prescribing practices for cystitis. This age group effect can be substantial for antibiotics that are commonly given to children, especially when the antibiotic is also used to treat a condition caused by *M. catarrhalis* (e.g. cefdinir and otitis media), but is less influential for antibiotics that are infrequently prescribed to children, such as ciprofloxacin and levofloxacin.

Overall, the proportion of avertable exposures is driven by pathogen, syndrome, and prescribing characteristics. Because our organisms of interest experience the vast majority of their antibiotic exposures as bystanders, the metric is largely dependent on the proportion of antibiotic use that is associated with the conditions affected in the given scenario. In a scenario where all antibiotic use for affected conditions is eliminated, this proportion would be equal to the proportion of avertable exposures for organisms exposed to antibiotics only during carriage (never pathogens). Being a causative pathogen for any of the affected conditions increases the proportion of avertable exposures. In contrast, being a causative pathogen for unaffected conditions increases unaffected antibiotic exposures and thus decreases the proportion of avertable exposures. For pathogens responsible for multiple syndromes, both factors may be at play. Finally, asymptomatic carriage prevalence is positively associated with both affected and unaffected antibiotic exposures, and, as a result, its overall effect on the proportion of avertable exposures is dependent on the antibiotic-species combination of interest.

## Discussion

We quantify the species-level impact of changes in antibiotic consumption as the proportion of antibiotic exposures experienced by common bacterial pathogens that could be averted under four hypothetical scenarios. In the scenario where unnecessary antibiotic use for all outpatient conditions is eliminated, we find that up to 48% of exposures (of *H. influenzae*, *E. coli*, and *P. aeruginosa* to azithromycin) could be avoided. In addition, impact of the intervention across antibiotics and species is widespread, with half of antibiotic-species pairs expected to experience a reduction in exposures of over one-fifth (22%). For conditions which require antibiotic treatment, such as UTIs, switching to antibiotics with decreased collateral damage to the microbiome, such as nitrofurantoin, may be an effective strategy for reducing antibiotic exposures across species.

In three out of four situations we assess, antibiotics are considered entirely unnecessary for some or all cases. In the future, this method may also be extended to measure net changes in exposures resulting from more nuanced scenarios where one antibiotic is substituted for another. We did not assess such changes for nitrofurantoin, as its effects on the microbiome outside of the bladder are thought to be minimal (*Stewardson et al., 2015*). The reductions in use presented here may be practically infeasible due to challenges including similar clinical presentation of viral and bacterial infections, laboratory processing times that prevent identification of causative pathogens during outpatient visits, individual considerations such as allergies (*Sakoulas et al., 2019*) or heightened risk of adverse events, misdiagnosis (*Filice et al., 2015*; *Tomas et al., 2015*), and patient-driven demand (*Vanden Eng et al., 2003*). Even if elimination of unnecessary use were fully realized, our results imply that the majority of species' antibiotic exposures occur in the context of 'necessary' antibiotic use. These findings underscore the importance of considering bystander effects and the need for a multi-pronged approach to programs aimed at controlling antibiotic resistance. Rapid diagnostics, diagnostics that accurately discriminate between bacterial and non-bacterial causes, patient education, and improved decision-making tools and other interventions to motivate changes in clinician behavior can enhance responsible antibiotic consumption and reduce unnecessary antibiotic use; these should be implemented simultaneously with prevention measures such as infection control, access to clean water and sanitation, safe sex interventions, and vaccination, which aim to reduce infection incidence and thus any antibiotic use.

At the time of this analysis, NHAMCS data from hospital outpatient departments was not available for 2015. In the 2010–2011 NAMCS/NHAMCS data, visits to hospital outpatient departments accounted for 36% of included sampled visits, but just 9% of all antibiotic mentions. Since the proportions of avertable exposures including and excluding these visits for 2010–2011 were similar for Scenario 1 (*Figure 1—figure supplement 3*), we assumed that the existing 2015 data without hospital outpatient department visits was representative of the NAMCS/NHAMCS population for the purposes of our analysis. Certain outpatient encounters are also outside the scope of NAMCS/NHAMCS, including federal facility visits, telephone contacts, house calls, long-term care stays, urgent care, retail clinics, and hospital discharge prescriptions.

Additional limitations of the included datasets and method for estimating antibiotic exposures have been previously enumerated (*Tedijanto et al., 2018*). Notably, the NAMCS/NHAMCS does not provide information to link medications with diagnoses, so we adopted a published tiered system to pair antibiotic use at each visit with the most likely indication (*Fleming-Dutra et al., 2016*). For visits with multiple diagnoses, all antibiotic use was attributed to the single most-likely indication. This method tends to overestimate antibiotic use for conditions for which antibiotics are almost always indicated, which may lead to clinically unusual diagnosis-treatment linkages. In the same way, antibiotic prescriptions are underestimated for diagnoses for which antibiotic use is not indicated, potentially leading to downward bias in avertable exposures for these conditions. Additionally, NAMCS/NHAMCS antibiotic use data does not include prescription details such as duration or dose. Future studies with more granular data may extend the methods presented here to account for such factors.

At the time of this analysis, Fleming-Dutra et al. remained the most up-to-date source of unnecessary antibiotic prescribing by condition. Recent work with slightly different methods found a lower proportion of inappropriate antibiotic prescriptions overall (23.2% compared to 30%) but did not report their estimates by condition (*Chua et al., 2019*). Although comparable estimates of unnecessary prescribing over time are unavailable, multiple studies have shown that declines in antibiotic use

from approximately 2011 to 2016 in the United States have been primarily due to pediatric prescribing, implying that, at least for adults, levels of unnecessary prescribing likely remained similar (*Durkin et al., 2018*; *King et al., 2020*; *Klevens et al., 2019*; *Olesen et al., 2018b*). We also assume that the proportion of unnecessary use is constant across all antibiotic use for the same condition as it is difficult to identify specific antibiotic prescriptions that were unwarranted without detailed chart review. A study among outpatients in the Veterans Affairs medical system found that among prescriptions for community-acquired pneumonia, sinusitis, and acute exacerbations of chronic bronchitis, the highest proportion of macrolide use was inappropriate (27%), followed by penicillins (22%) and quinolones (12%) (*Tobia et al., 2008*). Similar studies are needed to understand which antibiotics are frequently used inappropriately for other indications and settings.

Finally, it is important to note that the reduction in antibiotic exposures estimated here does not translate to the same reduction in the prevalence of resistance or in the morbidity and mortality attributable to resistance. Although population-level antibiotic consumption has been positively correlated with levels of antibiotic resistance, the impact of changing consumption is not well-understood and is likely to vary widely by antibiotic-species combination (or even antibiotic-strain combination) based on resistance mechanisms, fitness costs, and co-selection, among other factors (*Andersson and Hughes, 2010*; *Pouwels et al., 2017*). Antibiotic consumption is also highly heterogeneous and the impact of stewardship on resistance may depend on the affected patient populations (*Olesen et al., 2018a*). Additionally, in this analysis, we give each exposure equal weight. However, the selective pressure imposed by a single exposure depends on a number of variables, including pharmacokinetics, pharmacodynamics, distribution of bacteria across body sites, and bacterial population size. For example, we might expect that the probability of resistance scales with population size, and thus that an exposure received by an individual with higher bacterial load will have a larger impact on resistance. Further research, integrating knowledge from clinical, ecological, and evolutionary spheres, is needed to elucidate the relationship between antibiotic use and selective pressures and ultimately between use and resistance at the population level (*MacLean and San Millan, 2019*).

Reductions in antibiotic consumption are necessary to preserve the potency of these drugs. Quantifying changes in species-level exposures due to stewardship programs is one more step towards understanding how changes in antibiotic use correspond to antibiotic resistance. The methods presented here may be easily extended to incorporate other data sources, such as claims, or to assess more specific stewardship programs. We show that while improved prescribing practices have the potential to prevent antibiotic exposures experienced by bacterial species throughout the microbiome, complementary efforts to facilitate appropriate antibiotic consumption and decrease overall infection incidence are required to substantially avert exposures.

## Acknowledgements

We thank Dr. Lauri Hicks for her helpful comments on this manuscript.

## Additional information

### Competing interests

Marc Lipsitch: Reviewing editor, *eLife*. Yonatan H Grad: Has received consulting income from Merck and GlaxoSmithKline. The other author declares that no competing interests exist.

### Funding

| Funder | Grant reference number | Author |
| --- | --- | --- |
| National Institute of General Medical Sciences | U54GM088558 | Marc Lipsitch |
| National Institute of Allergy and Infectious Diseases | R01AI132606 | Yonatan H Grad |
| Centers for Disease Control and Prevention | CK000538-01 | Marc Lipsitch |

| Doris Duke Charitable Foundation | | Yonatan H Grad |
|---|---|---|
| National Institute of Allergy and Infectious Diseases | T32AI007535 | Christine Tedijanto |

The funders had no role in study design, data collection and interpretation, or the decision to submit the work for publication.

### Author contributions
Christine Tedijanto, Conceptualization, Formal analysis, Visualization, Methodology; Yonatan H Grad, Supervision, Funding acquisition, Methodology; Marc Lipsitch, Conceptualization, Supervision, Funding acquisition, Methodology

### Author ORCIDs
Christine Tedijanto ⑩ https://orcid.org/0000-0003-3403-5765
Yonatan H Grad ⑩ https://orcid.org/0000-0001-5646-1314
Marc Lipsitch ⑩ http://orcid.org/0000-0003-1504-9213

### Decision letter and Author response
Decision letter https://doi.org/10.7554/eLife.52307.sa1
Author response https://doi.org/10.7554/eLife.52307.sa2

## Additional files

### Supplementary files
• Transparent reporting form

### Data availability
Data from the 2015 National Ambulatory Medical Care Survey (NAMCS) and National Hospital Ambulatory Medical Care Survey (NHAMCS) are publicly available from the National Center for Health Statistics. This study also uses data from published literature, including the Human Microbiome Project and other studies summarized in Figure 1—source data 1.

The following previously published datasets were used:

| Author(s) | Year | Dataset title | Dataset URL | Database and Identifier |
|---|---|---|---|---|
| National Center for Health Statistics | 2015 | National Hospital Ambulatory Medical Care Survey | ftp://ftp.cdc.gov/pub/Health_Statistics/NCHS/dataset_documentation/nhamcs/stata | National Center for Heath Statistics, ED2015-stata |
| National Center for Health Statistics | 2015 | National Ambulatory Medical Care Survey | ftp://ftp.cdc.gov/pub/Health_Statistics/NCHS/dataset_documentation/namcs/stata | National Center for Heath Statistics, namcs2015-stata |

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
