## [Decision Letter]

**Acceptance summary:**

This manuscript provides a quantitative framework for assessing the impact of potential interventions to reduce unnecessary antibiotic use. It combines data on the prevalence of both infection-associated and colonizing bacterial populations with publicly available prescription data to assess the frequency of intentional vs. bystander antibiotic exposure in clinical practice. It then applies a modelling approach to understand how reduction in antibiotic prescriptions will affect overall antibiotic exposure across a variety of antibiotic-pathogen pairs. This extends previous work to provide further insights into the potential effectiveness of different antibiotic stewardship regimes.

**Decision letter after peer review:**

Thank you for submitting your article "Potential impact of outpatient stewardship interventions on antibiotic exposures of bacterial pathogens" for consideration by *eLife*. Your article has been reviewed by four peer reviewers, including Miles Davenport as the Reviewing Editor and Reviewer #1, and the evaluation has been overseen by a Reviewing Editor and Neil Ferguson as the Senior Editor. The following individuals involved in review of your submission have agreed to reveal their identity: Marc Bonten (Reviewer #3).

The reviewers have discussed the reviews with one another, and the Reviewing Editor has drafted this decision to help you prepare a revised submission.

Summary:

This manuscript looks at strategies to reduce antibiotic prescribing, and the role of 'bystander' exposure of commensals. The authors present a theoretical framework and nice thought experiment to quantify the potential benefits of changing antibiotic prescription in the outpatient setting. They have used "publicly available data to quantify potentially avertable exposures of bacterial species to commonly used antibiotics under hypothetical changes in prescribing practice." The four scenarios are relevant and "All scenarios demonstrate that stewardship interventions, facilitated by changes in clinician behaviour and improved diagnostics, have the opportunity to broadly reduce antibiotic exposures across a range of potential pathogens." The methodology builds upon previous work of the authors on the effects of antibiotics in the open population and the methods are clearly described. The limitations of the data regarding to infection coding and true antibiotic exposure and the uncertainty that any of the four strategies can be realized in real life are addressed appropriately. And despite the – somewhat – obvious conclusions ("less is more, but sometimes less more") this work presents a very thoughtful and interesting framework for future analyses.

The take home message is that exposure is reduced but would remain substantial, so outpatient stewardship alone is not enough.

Essential revisions:

1) The impact of changes prescribing strategy is interesting, but actually only relevant to the particular circumstance of bacteria that are associated with infection and either treated or not treated appropriately (see (a) below). This does not discount the importance of the insights – but in fact it feels that the manuscript presented numbers more than insights (see (b) below). Thus, the manuscript would be greatly strengthened and appeal to a much broader audience if the current results could be supplemented by some more intuitive explanations and ways to think about the problem.

a) While giving a percentage reduction is interesting, providing an intuition of how we might think about this and how it applies to different bacteria/drug pathogens combinations would be even more helpful. This is described to some extent in the Results section – but this was not an easy passage to follow unless you have a clear mental picture of pathogen/syndrome/antibiotic combinations, and could be explained much better in terms of 'classes' of pathogens. For example, if eliminating unnecessary use of penicillin for respiratory infection reduced overall penicillin prescriptions by 30%, then we can consider how this impact on three different classes of pathogen. (i) For unrelated pathogens (not specifically associated with respiratory infection), exposure is reduced by 30%. (ii) For pathogens that cause respiratory infection – but for which prescription is unnecessary and avoided by the strategy – they will have a >30% reduction (eg: overall reduction, + presence in infection * prescription rate (or some simple intuitive formula)). (iii) For pathogens that cause respiratory infection and for which penicillin is indicated, then they will have a lower reduction than 30% (ie: 30% – correct prescription rate).

This is not saying this issue is not described at all here (in formulas, and in the Results section) – just that it was not immediately clear that this fell out so easily, and the reader has to think about it. A flowchart-type figure of these classes and effects would make it very clear how the numbers in the tables were arrived at.

b) One thing that is not considered here is pathogen population size/density. For example, one might expect that the number of bacteria present in the microbiome may be many orders of magnitude lower than that present in the context of infection caused by the bacterium. Since the probability of resistance (and transmission of resistance) no doubt scales with population size, this seems important to discuss. This could be simply summarised as a 'ratio' even if not known precisely. That is, if the results were presented in the intuitive way described in (a) above, the issue of ratio could be introduced even without actual numbers of bacteria in microbiome/infection being known. For example, discussion of the impact of 'density' on the different scenarios above (eg: in scenario (ii) above, exposure per bacterium is reduced more than expected (because the treatments avoided are high density ones), whereas for scenario (iii) the reduction would be lower than calculated. (indeed, in the simple formula proposed above for a figure – the ratio of density in commensal carriage vs. infection could be added).

2) An additional paragraph on the relationship between antibiotic exposure and antibiotic resistance would be valuable. I realize the authors are avoiding that because it is not at all straightforward, but I think worth stating that a 46% exposure does not mean a 46% reduction in the frequency of resistance (let alone a 46% reduction in the morbidity/mortality associated with AMR). It might mean more or less; we have no idea. There is often a hidden assumption in the stewardship literature that if we can half exposure, we half the problem. The next important frontier is to figure out the relationship between the exposure and the resistance problem.

3) The four scenarios chosen are interesting and clinically relevant. However, they are all fairly different for one another in terms of clinical behavior change and potential implications. Therefore, it could be helpful to the reader if results for each scenario were presented separately in the results and discussion, with additional section tying all findings together and evaluating impact of all four scenarios together.

4) Additionally, for Scenario 1 it may be helpful to briefly define what is meant by unnecessary antibiotic use for readers unfamiliar with Fleming-Dutra et al. as it may be easy to confuse that with Scenario 2.

5) There is an issue of assuming "20/20 hindsight": the 'incorrect prescription' rate is based upon post-hoc analysis of the proportion of infections caused by a given pathogen. This information is not known to the clinician in advance. Therefore, there is no way to reach 'optimal targeting' in real-time prescribing. This should be discussed in more detail.

6) For acute sinusitis, it needs to be clarified that antibiotic treatment of acute bacterial sinusitis is currently guideline-recommended. Therefore, until guideline recommendations change, this scenario is unlikely to come to fruition. It may be helpful for the authors to consider a sensitivity analysis for acute sinusitis looking at (1) if only unnecessary use was eliminated (using Fleming-Dutra et al. definition) and (2) if all treatment was eliminated (current analysis). Some of this already exists in the supplement (Figure 1—figure supplement 2) and main text, so little additional analysis would be needed.

7) Consider being more explicit about codes/diagnoses are being considered in Scenarios 1, 2, and 4. Perhaps including a table in the supplement would be a nice way to summarize this.

8) The incorporation of sensitivity analyses and discussion of outpatient departments is unnecessary and may be confusing to readers. Many outpatient settings are excluded from NAMCS/NHAMCS. Just mention that in the methods and limitations.

---

## [Author Response]

Essential revisions:1) The impact of changes is prescribing strategy is interesting, but actually only relevant to the particular circumstance of bacteria that are associated with infection and either treated or not treated appropriately (see (a) below). This does not discount the importance of the insights – but in fact it feels that the manuscript presented numbers more than insights (see (b) below). Thus, the manuscript would be greatly strengthened and appeal to a much broader audience if the current results could be supplemented by some more intuitive explanations and ways to think about the problem.a) While giving a percentage reduction is interesting, providing an intuition of how we might think about this and how it applies to different bacteria/drug pathogens combinations would be even more helpful. This is described to some extent in the Results section – but this was not an easy passage to follow unless you have a clear mental picture of pathogen/syndrome/antibiotic combinations, and could be explained much better in terms of 'classes' of pathogens. For example, if eliminating unnecessary use of penicillin for respiratory infection reduced overall penicillin prescriptions by 30%, then we can consider how this impact on three different classes of pathogen. (i) For unrelated pathogens (not specifically associated with respiratory infection), exposure is reduced by 30%. (ii) For pathogens that cause respiratory infection – but for which prescription is unnecessary and avoided by the strategy – they will have a >30% reduction (eg: overall reduction, + presence in infection * prescription rate (or some simple intuitive formula)). (iii) For pathogens that cause respiratory infection and for which penicillin is indicated, then they will have a lower reduction than 30% (ie: 30% – correct prescription rate).This is not saying this issue is not described at all here (in formulas, and in the Results section) – just that it was not immediately clear that this fell out so easily, and the reader has to think about it. A flowchart-type figure of these classes and effects would make it very clear how the numbers in the tables were arrived at.

We thank the reviewers for their constructive feedback and agree that broader, more intuitive statements may be useful for a wider audience. Describing pathogen classes is slightly more complicated than outlined above because exposures depend not only on etiology for the condition of interest (e.g. respiratory infection) and carriage, but also on the propensity of an organism to cause other infections. In addition, whether antibiotics are or are not indicated does not depend entirely on the causative pathogen. These complexities make it difficult to put forth simple formulas for the proportion of avertable exposures by pathogen “class”, but we have included a set of more general statements at the end of the Results section. We hope that this approach will provide readers with the desired intuition.

We have also added an additional source data file (Figure 1—source data 3) including tables of minimally processed data inputs to help facilitate better understanding of the underlying calculations and enable further exploration/manipulation by readers.

b) One thing that is not considered here is pathogen population size/density. For example, one might expect that the number of bacteria present in the microbiome may be many orders of magnitude lower than that present in the context of infection caused by the bacterium. Since the probability of resistance (and transmission of resistance) no doubt scales with population size, this seems important to discuss. This could be simply summarised as a 'ratio' even if not known precisely. That is, if the results were presented in the intuitive way described in (a) above, the issue of ratio could be introduced even without actual numbers of bacteria in microbiome/infection being known. For example, discussion of the impact of 'density' on the different scenarios above (eg: in scenario (ii) above, exposure per bacterium is reduced more than expected (because the treatments avoided are high density ones), whereas for scenario (iii) the reduction would be lower than calculated. (indeed, in the simple formula proposed above for a figure – the ratio of density in commensal carriage vs. infection could be added).

We thank the reviewers for their comment and acknowledge that bacterial density could play a role in propagation of resistance. However, we do not believe there is enough data to support the idea that an exposure during treatment is more important for resistance than an exposure experienced during colonization. There are also reasons to believe the opposite – for example, infections reflect bacteria that found their way into normally sterile sites, and resistant strains or their genes may find it harder to transmit from these sites (e.g. the bloodstream is thought to be a “dead end for evolution” (Levin and Bull, 1994). There is insufficient evidence to conclude whether infection is associated with higher loads of the causative pathogen throughout the body. In addition, many other variables beyond density may modulate the interaction between antibiotics and infecting and colonizing populations of bacteria. For example, pharmacokinetics, pharmacodynamics, the nature of the particular infection (e.g. abscess that is hard for antibiotics to penetrate) and the non-uniform distribution of bacteria across body sites may also play a role.

We have added several lines to the Discussion section to acknowledge that population size/density, along with other factors mentioned above, may impact our findings.

2) An additional paragraph on the relationship between antibiotic exposure and antibiotic resistance would be valuable. I realize the authors are avoiding that because it is not at all straightforward, but I think worth stating that a 46% exposure does not mean a 46% reduction in the frequency of resistance (let alone a 46% reduction in the morbidity/mortality associated with AMR). It might mean more or less; we have no idea. There is often a hidden assumption in the stewardship literature that if we can half exposure, we half the problem. The next important frontier is to figure out the relationship between the exposure and the resistance problem.

We thank the reviewers for raising this important point. We have added a paragraph on this issue to the Discussion section.

3) The four scenarios chosen are interesting and clinically relevant. However, they are all fairly different for one another in terms of clinical behavior change and potential implications. Therefore, it could be helpful to the reader if results for each scenario were presented separately in the results and discussion, with additional section tying all findings together and evaluating impact of all four scenarios together.

As recommended, the Results section has been separated by scenario. We hope this will help improve clarity for readers. Most of the Discussion section is broadly applicable and serves to tie the scenarios together.

4) Additionally, for Scenario 1 it may be helpful to briefly define what is meant by unnecessary antibiotic use for readers unfamiliar with Fleming-Dutra et al. as it may be easy to confuse that with Scenario 2.

We thank the reviewers for raising this potential point of confusion. A brief paragraph has been added to the Materials and methods section to define unnecessary antibiotic use as estimated by Fleming-Dutra et al.

5) There is an issue of assuming "20/20 hindsight": the 'incorrect prescription' rate is based upon post-hoc analysis of the proportion of infections caused by a given pathogen. This information is not known to the clinician in advance. Therefore, there is no way to reach 'optimal targeting' in real-time prescribing. This should be discussed in more detail.

We thank the reviewers for this important point. We have added “laboratory processing times that prevent identification of causative pathogens during outpatient visits” to the list of issues that preclude optimal prescribing (Discussion section); another related point that is already listed is “similar clinical presentation of viral and bacterial infections”. We also include rapid diagnostics as a key tool for reducing unnecessary antibiotic use (Discussion section).

6) For acute sinusitis, it needs to be clarified that antibiotic treatment of acute bacterial sinusitis is currently guideline-recommended. Therefore, until guideline recommendations change, this scenario is unlikely to come to fruition. It may be helpful for the authors to consider a sensitivity analysis for acute sinusitis looking at (1) if only unnecessary use was eliminated (using Fleming-Dutra et al. definition) and (2) if all treatment was eliminated (current analysis). Some of this already exists in the supplement (Figure 1—figure supplement 2) and main text, so little additional analysis would be needed.

We have added this analysis as a third scenario in Figure 1—figure supplement 2 and clarified that antibiotic treatment of acute bacterial sinusitis is currently guideline-recommended in the text (subsection “Scenarios of interest”). Based on the Fleming-Dutra et al. paper, we estimate that 18% (0-19 years old), 100% (20-64 years old), and 34% (>64 years old) of antibiotic prescriptions given for acute sinusitis are unnecessary. Note that these estimates do not align directly with the numbers from the Fleming-Dutra paper (9% for 0-19 years old, 51% for 20-64 years old, and 16% for >64 years old) because the Fleming-Dutra paper groups together prescriptions for acute and chronic sinusitis. Although chronic sinusitis arises from multiple pathophysiologic processes, antibiotics may be indicated for chronic sinusitis patients with acute exacerbations and/or signs of infection. Thus, we back-calculate the modified proportions of unnecessary prescriptions for acute sinusitis alone under the simplifying assumption that all prescriptions for chronic sinusitis are necessary (see Figure 1—source data 2 for more details).

7) Consider being more explicit about codes/diagnoses are being considered in scenarios 1, 2, and 4. Perhaps including a table in the supplement would be a nice way to summarize this.

Explicit ICD-9-CM codes for each condition in Scenarios 1 and 2 can be found in eTable 2 of Fleming-Dutra et al., 2016. We describe in the text that we have directly applied their coding methodology: “We used methodology developed by Fleming-Dutra and colleagues, and applied in other studies, to group diagnosis codes into conditions…” (subsection “Data sources and methodology”). Codes for Scenario 4 are written out in the text in the Materials and methods section.

8) The incorporation of sensitivity analyses and discussion of outpatient departments is unnecessary and may be confusing to readers. Many outpatient settings are excluded from NAMCS/NHAMCS. Just mention that in the methods and limitations.

We apologize for any confusion this sensitivity analysis and discussion caused. The purpose of the sensitivity analysis in Figure 1—figure supplement 3 was not explicitly to explore the inclusion/exclusion of specific outpatient departments, but rather to explore validity of using the more recent 2015 data which is missing hospital outpatient department visits (over one-third of all sampled visits in the 2010-2011 data). Because we observed that the 2010-2011 results were similar with and without this segment of visits, we assumed the antibiotic use and diagnosis data from 2015 were representative of the entire NAMCS/NHAMCS population for the purposes of our metric.

To simplify, we have maintained only Scenario 1 in the sensitivity analysis and have added more detail in the Discussion section to explain this Figure supplement. We have also briefly mentioned the limitation of missing outpatient settings in NAMCS/NHAMCS (Discussion section).